

# BNC1 inhibits the development and progression of gastric cancer by regulating the CCL20/JAK-STAT axis

Lixin Liu[1,2], Li Xiong[3], Hong Peng[4], Qin Deng[5], Kang Liu[2] and Shusen Xia[1,6]

[1] The Second Department of Gastrointestinal Surgery, The Affiliated Hospital of the North Sichuan Medical College, Nanchong, Sichuan, China
[2] Institute of Tissue Engineering and Stem Cell, Beijing Anzhen Nanchong Hospital of Capital Medical University, Nanchong Central Hospital, The Second Clinical Medical College of North Sichuan Medical College, Nanchong, Sichuan, China
[3] Department of Clinical Laboratory, People's Hospital of Leshan, Leshan, Sichuan, China
[4] Department of Anorectal Surgery, Nanchong Central Hospital, The Second clinical Medical College, North Sichuan Medical College, Nanchong, Sichuan, China
[5] School of Clinical Medicine, Guizhou Medical University, Guiyang, Guizhou, China
[6] Department of Gastrointestinal Surgery, Clinical Medical College and the First Affiliated Hospital of Chengdu Medical College, Chengdu, Sichuan, China

## ABSTRACT

The role of basonuclin 1 (BNC1), a zinc finger protein-specific transcription factor, in gastric cancer remains unclear. In this study, BNC1 was downregulated in gastric cancer and functioned as a tumor suppressor. Through integrative analyses of transcriptome sequencing and functional assays, C-C motif chemokine ligand 20 (CCL20) was identified as a direct downstream target of BNC1. Overexpression of BNC1 inhibited the proliferation, migration, and invasion of gastric cancer cells both *in vitro* and *in vivo*. Mechanistically, BNC1 suppresses CCL20 expression by binding to its promoter, leading to reduced activation of the JAK-STAT signaling pathway and promoting apoptosis in gastric cancer cells. These findings highlight the pivotal role of BNC1 in gastric cancer progression and suggest that targeting BNC1 and its downstream pathways could serve as a potential therapeutic strategy.

## INTRODUCTION

Gastric cancer (GC) remains a significant global health concern, with over 968,000 new GC cases and approximately 660,000 deaths reported in 2022, making it the fifth leading cause of cancer mortality worldwide (*Bray et al., 2024*). Though gastric cancer rates are declining in China, the country has the highest gastric cancer rates in the world, accounting for 44% of total worldwide cases (*He et al., 2024a*). Recent advancements have been made in gastric cancer diagnosis and treatment in China; however, a substantial gap in disease burden and patient outcomes persists compared to developed nations (*He et al., 2024b*). Emerging evidence has highlighted the critical role of transcription factors in tumor proliferation and metastasis (*Yang et al., 2020*; *Zhu, Tang & Wu, 2020*). Identifying transcription factor

Corresponding authors
Kang Liu, liukang@nsmc.edu.cn
Shusen Xia, xsx0639@163.com

targets with therapeutic potential is crucial for developing novel strategies for the treatment of gastric cancer.

Basonuclin 1 (BNC1) is a zinc finger protein-specific transcription factor that exhibits a high degree of evolutionary conservation, underscoring its fundamental biological importance. Initially identified *in vitro* using cultured human epidermal keratinocytes, BNC1 has been implicated in the regulation of keratinocyte proliferation and differentiation, as well as in germ cell development (*Romano et al., 2004*; *Wu et al., 2016*; *Li et al., 2021*). BNC1's expression patterns and functional roles suggest that it may be critical in maintaining cellular homeostasis in specific tissues. In recent years, the role of BNC1 in cancer has garnered increased attention. Elevated levels of BNC1 expression have been reported in esophageal squamous cell carcinoma and basal cell carcinoma. In these malignancies, BNC1 is posited to act as an oncogene, contributing to processes such as cellular proliferation, invasion, and metastasis (*Cui et al., 2004*; *Xiong et al., 2023*). Conversely, studies have documented reduced expression of BNC1 in renal cell carcinoma and hepatocellular carcinoma, implying a potential tumor suppressor function (*Morris et al., 2010*; *Wu et al., 2016*). Although these findings provide valuable insights, there are currently no reported studies investigating the role of BNC1 in gastric cancer. Elucidating whether BNC1 functions as an oncogene or tumor suppressor in gastric cancer could enhance our understanding of disease mechanisms and potentially lead to the development of targeted interventions.

Transcription factors (TFs) bind specific DNA sequences and play pivotal roles in the transcriptional regulation of gene expression. This binding leads to the direct or indirect transcriptional activation of downstream genes by facilitating the action of RNA polymerase on promoter sequences (*Lambert et al., 2018*; *Soutourina, 2018*). Additionally, TFs can suppress transcription by recruiting repressors or by interfering with the binding of other transcription factors to promoters (*Moody et al., 2005*; *Reinke, Xu & Cheng, 2012*). In the current study, both *in vivo* and *in vitro* experiments confirmed that BNC1 acts as a tumor suppressor in gastric cancer cells. To investigate the molecular mechanisms by which BNC1 exerts its tumor-suppressive effects, the study employed an integrative approach combining transcriptome sequencing, bioinformatics analyses, and cellular functional assays to identify and validate C-C motif chemokine ligand 20 (CCL20) as a downstream target gene of BNC1.

Recent research indicates that cytokines are significantly upregulated in various cancers, and that they may be closely associated with cancer growth and progression within the tumor microenvironment (TME). This upregulation is implicated in mechanisms such as the evasion of the immune response, the promotion of angiogenesis, and the enhancement of tumor invasiveness (*Bell et al., 1999*; *Hanahan & Coussens, 2012*; *Tian et al., 2024*). Notably, studies have found that CCL20 plays a crucial role in the gastric tumor microenvironment and serves as a potential biomarker for GC (*Liu et al., 2023*). The current study further validates the oncogenic role of CCL20 in gastric cancer, underscoring its significance in the disease's pathology and highlighting its potential as a therapeutic target.

This study identifies BNC1 as a novel tumor suppressor and proposes CCL20 as an oncogene in gastric cancer. Mechanistically, BNC1 binds to the promoter region of CCL20, negatively regulating its expression. This suppression of CCL20 reduces activation of the CCL20-mediated JAK-STAT signaling pathway, thereby inhibiting tumor progression.

## MATERIALS & METHODS

### Clinical samples

Authorization for the procurement and use of patient specimens for this study was received from the Ethics Committee of The Affiliated Hospital of North Sichuan Medical College (Approval Number: 2024ER463-1). This investigation was exempted from obtaining informed consent as it solely entailed immunohistochemical (IHC) examination of archived pathological samples and used anonymized data extracted from the hospital's electronic medical records and databases, ensuring minimal risk to participants. A total of 61 matched pairs of gastric carcinoma tissues and corresponding adjacent normal tissues were acquired for IHC analysis. The pathological characteristics of these 61 gastric cancer cases are detailed in Table S1.

### Cell lines and cell culture

The GES-1 normal human gastric epithelial cell line was obtained from GuangZhou Jennio Biotech Company. The GC cell lines HGC-27, SNU-1, and AGS were sourced from Shanghai Min Jin Biotechnology Company. The MKN-28 and MGC-803 GC cell lines were acquired from cells maintained at the Institute of Tissue Engineering and Stem Cells, Nanchong Central Hospital. For cell culture, GES-1, MKN-28, MGC-803, and SNU-1 cells were maintained in RPMI-1640 medium (Gibco). HGC-27 cells were cultured in Dulbecco's Modified Eagle Medium (DMEM; Gibco), and AGS cells were cultured in F-12K medium (Gibco). All media were supplemented with 10% fetal bovine serum (FBS), 100 U/mL penicillin, and 100 μg/mL streptomycin. Cells were incubated at 37 °C in a humidified atmosphere containing 5% $CO_2$. To ensure the authenticity of the cell lines and prevent cross-contamination, short tandem repeat (STR) profiling was performed on all cell lines prior to experimentation.

### siRNAs and transfection

The small interfering RNAs (siRNAs) targeting CCL20 (si-CCL20#1 through si-CCL20#4) and the negative control siRNA (si-NC) were procured from GenePharma. Transfections were performed using Lipofectamine 2000 reagent (Invitrogen) following the manufacturer's protocol. Detailed sequences of the siRNAs are provided in Table S2. The experimental procedures were conducted in strict accordance with the product manuals provided by GenePharma (https://www.genepharma.com/en/).

### Quantitative RT-PCR

Forty-eight hours post-transfection, or upon establishment of stable GC cell lines, total RNA was extracted using the FastPure Cell/Tissue Total RNA Isolation Kit (Vazyme Biotech Co., Ltd). Genomic DNA was removed and complementary DNA (cDNA)

synthesis was performed with the HiScript III 1st Strand cDNA Synthesis Kit (+gDNA Wiper) from the same supplier. Quantitative PCR amplification of the cDNA was conducted using the ChamQ Universal SYBR qPCR Master Mix (Vazyme Biotech Co., Ltd; https://www.vazyme.com/). Primer sequences used for qPCR were synthesized by Sangon Biotech (Shanghai, China) and are detailed in Table S3. Relative gene expression levels were determined using the $2^{-\Delta\Delta Ct}$ method, with GAPDH (glyceraldehyde-3-phosphate dehydrogenase) serving as the internal control.

## ELISA validation

The stable BNC1-overexpressing MKN-28 and AGS cell lines were used to perform the ELISA analysis using the Human MIP-3α/CCL20 (C-C motif chemokine ligand 20) QuickTest ELISA Kit (QT-EH0231; Wuhan Fine Biotech Co., Ltd). Assays and analyses were conducted according to the manufacturer's protocol.

## Stable cell line construction

Lentiviral vectors engineered to overexpress BNC1 and confer puromycin resistance were procured from Shanghai GenePharma Co., Ltd. Prior to establishing stable cell lines, the minimum lethal concentration (MLC) of puromycin was determined by treating GC cells (MKN-28, MGC-803, and AGS) with increasing concentrations of puromycin (ranging from 0.5 to 5.0 μg/mL) over a three-day period. The MLC was identified as the lowest concentration of puromycin that resulted in complete cell death within this timeframe, which was found to be 1.0 μg/mL. Following transduction, the culture medium was refreshed with 10% FBS 24 h post-infection. After 72 h, selection commenced by introducing 10% FBS medium supplemented with 1.0 μg/mL puromycin. Stable gastric cancer cell lines overexpressing BNC1 were established within one week. The specific sequences used in the lentiviral constructs are detailed in Table S2.

## Immunohistochemistry

IHC analysis was performed on 61 pairs of formalin-fixed, paraffin-embedded GC tissues from clinical sources and four pairs of paraffin-embedded subcutaneous GC tissue sections from animal experiments. The primary antibodies used were: BNC1 polyclonal antibody (DF8812, 1:150; Jiangsu Family Biology Research Center, Ltd), CCL20 polyclonal antibody (DF2238, 1:100; Jiangsu Family Biology Research Center, Ltd), Ki-67 (ready-to-use; Fuzhou Maxin Biotechnology Development Co., Ltd), and Her-2 polyclonal antibody (FNab03833; Wuhan Fine Biotech Co., Ltd). Two pathologists independently evaluated the IHC staining using a scoring system that combined staining intensity and the percentage of positive cells, with the final score being the product of these two parameters, ranging from zero to nine points. The pathologists were blinded to both the clinical data of the specimens and the specific antibodies used during their assessments. The detailed scoring criteria were as follows: the proportion of positive cells was rated as 0 (0–25%), 1 (25–50%), 2 (50–75%), or 3 (greater than 75%); staining intensity was categorized as 0 (no staining), 1 (weak staining, light yellow), 2 (moderate staining, yellow), and 3 (strong staining, brown). The scoring results are presented in Tables S1 and S5.

## Colony formation assays

GC cells were harvested during the logarithmic growth phase and prepared as a single-cell suspension. Cells were seeded into six-well culture plates at a density of 1,000 cells per well, with each well containing two mL of culture medium supplemented with 10% FBS. The cultures were maintained at 37 °C in a humidified atmosphere with 5% $CO_2$, and the medium was refreshed every three days. After an incubation period of 14 days, allowing for colony development, the cells were gently washed twice with phosphate-buffered saline (PBS) to remove residual medium. Next, the colonies were fixed by adding one mL of 20% methanol to each well and incubating at room temperature for 30 min. Following fixation, the methanol was aspirated, and one mL of 0.2% crystal violet staining solution was added to each well with a staining duration of three minutes at room temperature. Excess stain was removed by rinsing the wells gently with distilled water, and the plates were air-dried at room temperature. For colony quantification, images of the stained colonies were captured using a high-resolution digital camera. The acquired images were analyzed using ImageJ software (National Institutes of Health, Bethesda, MD, USA).

## Dual luciferase reporter assays

To study the influence of BNC1 on CCL20 promoter activity, the researchers engineered the overexpression construct pCDNA3.1-BNC1 and the reporter plasmid pGL3-CCL20pro. AGS cells were seeded into 24-well plates at a density of $5 \times 10^4$ cells per well and cultured overnight in F12K medium supplemented with 10% fetal bovine serum at 37 °C in a humidified atmosphere containing 5% $CO_2$. The following day, cells were co-transfected with 800 ng of pGL3-CCL20pro and 200 ng of either pCDNA3.1 (empty vector) or pCDNA3.1-BNC1 using Lipofectamine 2000 (Thermo Fisher Scientific) according to the manufacturer's protocol. After 48 h of transfection, cells were lysed with 100 µL of Passive Lysis Buffer (Promega) per well. Next, 20 µL of the lysate was mixed with 100 µL of Luciferase Assay Reagent II to measure Firefly luciferase activity, followed by the addition of 100 µL of Stop & Glo Reagent to assess Renilla luciferase activity using the Dual-Luciferase Reporter Assay System (Promega) as per the manufacturer's instructions. Luminescence was quantified with a GloMax 20/20 luminometer (Promega). Firefly luciferase activity was normalized to Renilla luciferase activity to control for transfection efficiency.

## Western blot analysis

Proteins were extracted using RIPA lysis buffer supplemented with protease inhibitors, PMSF, and phosphatase inhibitors. The extracted proteins were separated on a 10% SDS-PAGE gel prepared using the Omni-Easy™ One-Step PAGE Gel Preparation Kit (Epizyme Biotech). Following electrophoresis, proteins were transferred onto a polyvinylidene fluoride (PVDF) membrane. The membrane was blocked with 5% nonfat dry milk for one hour at room temperature and then incubated overnight at 4 °C with the following primary antibodies: BNC1 (PA5-66883, 1:2000; Invitrogen), JAK2 (R013499, 1:500; Shanghai Epizyme Biomedicine Technology Co., Ltd), STAT3 (10253-2-AP, 1:2000; Proteintech), p-STAT3 (28945-1-AP, 1:1000; Proteintech), BCL-2 (12789-1-AP, 1:2000; Proteintech), and BAX (50599-2-Ig, 1:2000; Proteintech). After four washes with phosphate-buffered

solution with Tween 20 (PBS-T), the membrane was incubated with horseradish peroxidase (HRP)-conjugated goat anti-rabbit IgG (1:1000) or HRP-conjugated goat anti-mouse IgG (1:1000). Signal detection was performed using an enhanced chemiluminescence (ECL) reagent (Bio-Rad, Hercules, CA, USA), and images were captured using a gel imaging system or developed on X-ray film in a darkroom. Quantitative analysis of band intensity was conducted using ImageJ software (National Institutes of Health).

### Flow cytometry

To evaluate apoptosis, $2 \times 10^5$ MKN-28 and AGS cells overexpressing BNC1 were harvested. The cells were centrifuged at $300\times$ g for five minutes, the supernatant was removed, and the pellet was washed twice with PBS. After the final wash, cells were resuspended in 500 μL of $1 \times$ Annexin V Binding Buffer. Next, 5 μL of Annexin V-FITC and five μL of propidium iodide (PI, 50 μg/mL) were added. The mixture was gently vortexed and incubated in the dark at room temperature for 20 min. Flow cytometry analysis was then performed immediately.

For cell cycle analysis, approximately $4 \times 10^5$ MKN-28 and AGS cells stably overexpressing BNC1 were collected. The cells were washed twice with cold PBS and resuspended in 0.3 mL of PBS. To this suspension, 1.2 mL of pre-chilled absolute ethanol ($-20\,°C$) was added while vortexing, and the cells were fixed at $-20\,°C$ for at least one hour or overnight. Post-fixation, cells were centrifuged at $300\times$ g for five minutes, the supernatant was discarded, and the pellet was resuspended in one mL of PBS and incubated at room temperature for 15 min. After another centrifugation and supernatant removal, 100 μL of RNase A reagent was added to fully resuspend the cells, followed by a 30-minute incubation in a $37\,°C$ water bath. Next, 400 μL of PI reagent (50 μg/mL) was added, mixed thoroughly, and the cells were incubated in the dark at $2–8\,°C$ for 30 min. Stained cells were immediately analyzed using a FACSCalibur flow cytometer (BD Biosciences), and data were processed with FlowJo software (BD Biosciences). All reagents used in these procedures were obtained from Elabscience Biotechnology Co., Ltd.

### Xenograft model

Athymic nude mice were purchased from Chengdu GemPharmatech Biotechnology Co., Ltd. The mice were housed and cared for by trained professionals at the Animal Experimentation Center of North Sichuan Medical College. The animals were kept in a controlled environment with a temperature of $22 \pm 2\,°C$ and a 12-hour light/dark cycle. Each cage housed three to five mice with ad libitum access to food and water. A total of 20 animals were used for the subcutaneous tumor model. A random sequence generation method was used to allocate the animals into control and experimental groups with 10 mice in each group. All experimental mice were six-week-old male nude mice weighing approximately 20 g. After one week of acclimatization, the athymic nude mice were injected subcutaneously with $2 \times 10^6$ AGS/NC or AGS/OE-BNC1 cells. Mouse weight and tumor volume were monitored every three days. Measurements were conducted by two individuals using the same caliper and weighing scale, with each researcher blinded to the other's measurements. The final values were recorded as the average of the two

independent measurements. Tumor size was calculated using the following formula: tumor volume (mm$^3$) = width × length × (width + length) × 0.5. Animals were euthanized *via* intraperitoneal injection of euthanasia agents following predefined humane endpoints or experimental termination criteria. Humane endpoints included: (1) a 25% reduction in body weight, (2) complete loss of appetite for 24 h, or (3) inability to stand for 24 h due to weakness or an extreme reluctance to stand. Animals were also euthanized at experimental termination points, which included the end of the observation period (21 days post-injection) or a maximum of eight weeks post-treatment. Tumors were isolated upon euthanasia for photographing, IHC analysis, and hematoxylin and eosin (H&E) staining. If the subcutaneous tumor model failed to establish successfully or if animal mortality occurred during the experiment, both resulting in measurement failure, the corresponding data was excluded from the analysis. Animal studies were carried out according to the protocols approved by the ethics committee of the Institutional Animal Care and Use Committee of the North Sichuan Medical College (IACUC approval number: NSMC2024102).

## Migration and invasion assays

The cells from each group were harvested through trypsinization, subjected to centrifugation, and rinsed twice with serum-free medium to eliminate any remaining serum. For the migration assay, cells were resuspended in serum-free medium at a concentration of $2 \times 10^5$ cells/mL. A 100 μL aliquot of this suspension was added to the upper chamber of each Transwell insert, while the lower chamber was filled with medium supplemented with 10% serum. After 48 h of incubation, the upper chambers were removed, and cells on the membrane's upper surface were gently wiped away using a moistened cotton swab. The membranes were then fixed in 20% methanol and stained with 0.1% crystal violet for ten minutes. Cells that had migrated to the lower surface were visualized, photographed, and counted under a light microscope at 100× magnification. For the invasion assay, the procedure mirrored that of the migration assay with the following modifications: the upper chambers were pre-coated with Matrigel, and the cells were resuspended at a density of $4 \times 10^5$ cells/mL. All subsequent steps were conducted as described for the migration assay. Quantification of migrated and invaded cells was performed using ImageJ software.

## Wound healing test

Cells stably expressing the lentiviral construct were seeded into six-well plates. After ensuring complete adherence, a sterile pipette tip was utilized to create uniform linear scratches in each well. Wound closure was monitored by capturing images at zero and 72 h using an inverted microscope. The migration rate was determined using the following formula: Scratch healing rate (%) = [(initial scratch width − scratch width at 72 h)/initial scratch width] × 100.

## Cell counting kit-8 method

Cells stably expressing the lentiviral construct were harvested, enzymatically dissociated, and resuspended in culture medium. Following cell counting, 3,000 cells per well were

plated into 96-well plates. Cell proliferation was evaluated at 24, 48, 72, and 96 h using the Cell Counting Kit-8 (CCK-8) assay (Vazyme Biotech Co., Ltd) by adding 10 μL of CCK-8 reagent to each well and incubating for one hour. Absorbance at 450 nm was measured with a microplate reader to determine cell viability.

## Statistical analysis

Statistical analyses were conducted using GraphPad Prism version 9.0. Data were expressed as the mean ± standard deviation (SD) from a minimum of three independent experiments. Group comparisons were performed using one-way analysis of variance (ANOVA) while differences between two groups were assessed with Student's $t$-test. Statistical significance was defined as $P < 0.05$, with significance levels denoted as follows: $*P < 0.05$, $**P < 0.01$, $***P < 0.001$, and $****P < 0.0001$.

## RESULTS

### Clinical relevance analysis for basonuclin 1

To investigate the role of the novel transcription factor basonuclin 1 (BNC1) in gastric cancer, IHC staining was performed on GC tissues and adjacent normal tissues randomly collected from 61 patients. BNC1 expression was predominantly nuclear and was significantly lower in cancer tissues compared to adjacent normal tissues (Figs. 1A, 1B). Additional analyses using patient clinical information (Table S1) revealed no significant correlation between BNC1 expression and tumor differentiation or patient gender. However, a significant relationship was observed between BNC1 expression and both lymph node metastasis and pathological stage (Figs. 1C–1E and Figs. S1A, S1B). BNC1 expression decreased progressively with increasing numbers of metastatic lymph nodes and advancing pathological stages. These findings suggest that reduced BNC1 expression is associated with tumor progression and metastasis in gastric cancer. Consequently, BNC1 may serve as a potential molecular marker for clinical diagnosis and is closely linked to the development and progression of gastric cancer.

### Overexpression of BNC1 suppresses development of gastric cancer cells

To investigate the functional role of BNC1 in gastric cancer cells, its expression was assessed in the normal gastric epithelial cell line GES-1 and five gastric cancer cell lines—HGC-27, SNU-1, MGC-803, MKN-28, and AGS—using quantitative real-time PCR (qRT-PCR) and Western blot analyses (Figs. 2A–2C). Notably, the SNU-1 cell line is not suitable for conducting wound healing assays. As such, MGC-803 and AGS cell lines, which showed the most significant differences in BNC1 expression, were initially chosen for functional assays. However, modulation of BNC1 in the MGC-803 cell line did not result in significant biological changes (Figs. S2A, S2B). Therefore, MKN-28 and AGS cell lines were used for further experiments. Using a lentiviral system, stable BNC1-overexpressing MKN-28 and AGS cell lines were successfully established and confirmed by qRT-PCR and Western blot analyses (Figs. S2C–S2E). Compared to control cells, BNC1 overexpression significantly inhibited cell proliferation (Figs. 2D–2F). Transwell assays and wound healing assays

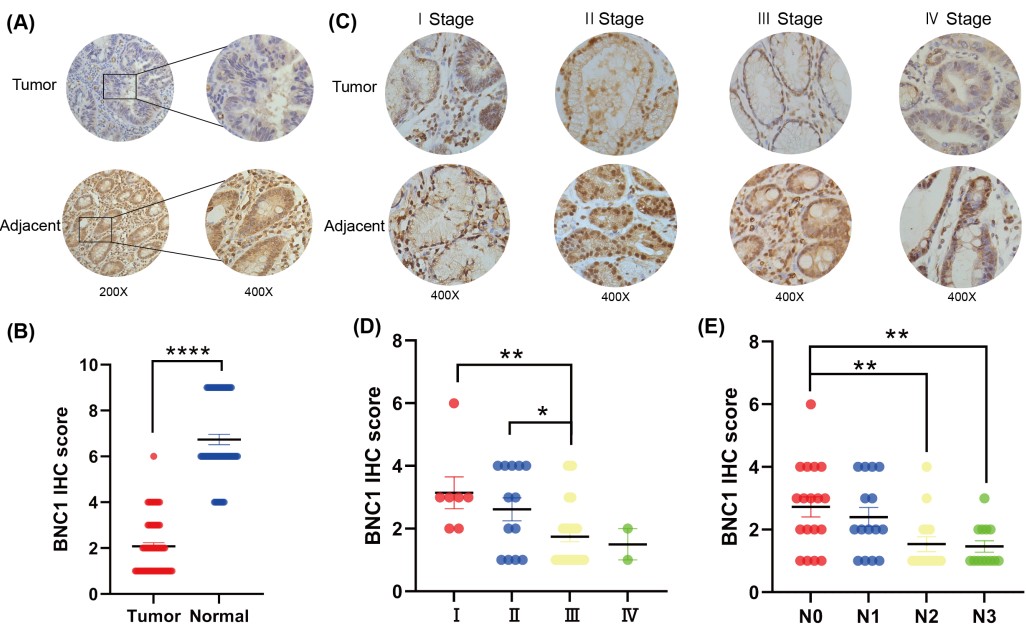

**Figure 1  Clinical expression of BNC1 in GC.** (A–B) Representative images and quantification of BNC1 IHC staining in gastric tumor tissues and adjacent normal tissues ($n = 61$; magnification $200\times$ and $400\times$). (C–D) Representative images and quantification of BNC1 IHC staining in gastric cancer tissues of different pathological stages show that BNC1 expression is progressively reduced with increasing tumor stage ($n = 61$; magnification $400\times$). (E) Quantification of BNC1 IHC staining in gastric cancer tissues reveals a further decrease in BNC1 expression as lymph node metastasis increases.

demonstrated that the migratory and invasive capabilities of BNC1-overexpressing cells were markedly reduced (Figs. 2G–2J). Flow cytometry revealed that BNC1 overexpression induced G1 phase cell cycle arrest (Fig. 2K, Fig. 2L). Additionally, apoptosis assays indicated that BNC1 overexpression resulted in a significant increase in apoptotic cells (Figs. 2M–2P). These findings suggest that BNC1 acts as a tumor suppressor in gastric cancer cells.

## BNC1 suppresses proliferation and metastasis of gastric cancer cells *in vivo*

To further investigate the function of BNC1 in nude mice, an AGS cell stable strain that overexpressed BNC1 using lentivirus was constructed. Subcutaneous tumor model experiments were carried out, and the results showed that, after overexpressing BNC1, tumor volume and tumor weight were significantly reduced (Figs. 3A–3D) and the nuclear-cytoplasmic ratio of tumors was also significantly reduced (Fig. 3E). Further IHC analysis was conducted to assess the proliferation and expression of metastasis markers. The results showed that the expression levels of Ki-67 and Her-2 were markedly lower in the BNC1 overexpression group compared to the control group (Figs. 3F–3H). These findings corroborate the *in vitro* results and imply that BNC1 overexpression suppresses tumor growth and metastasis *in vivo*.

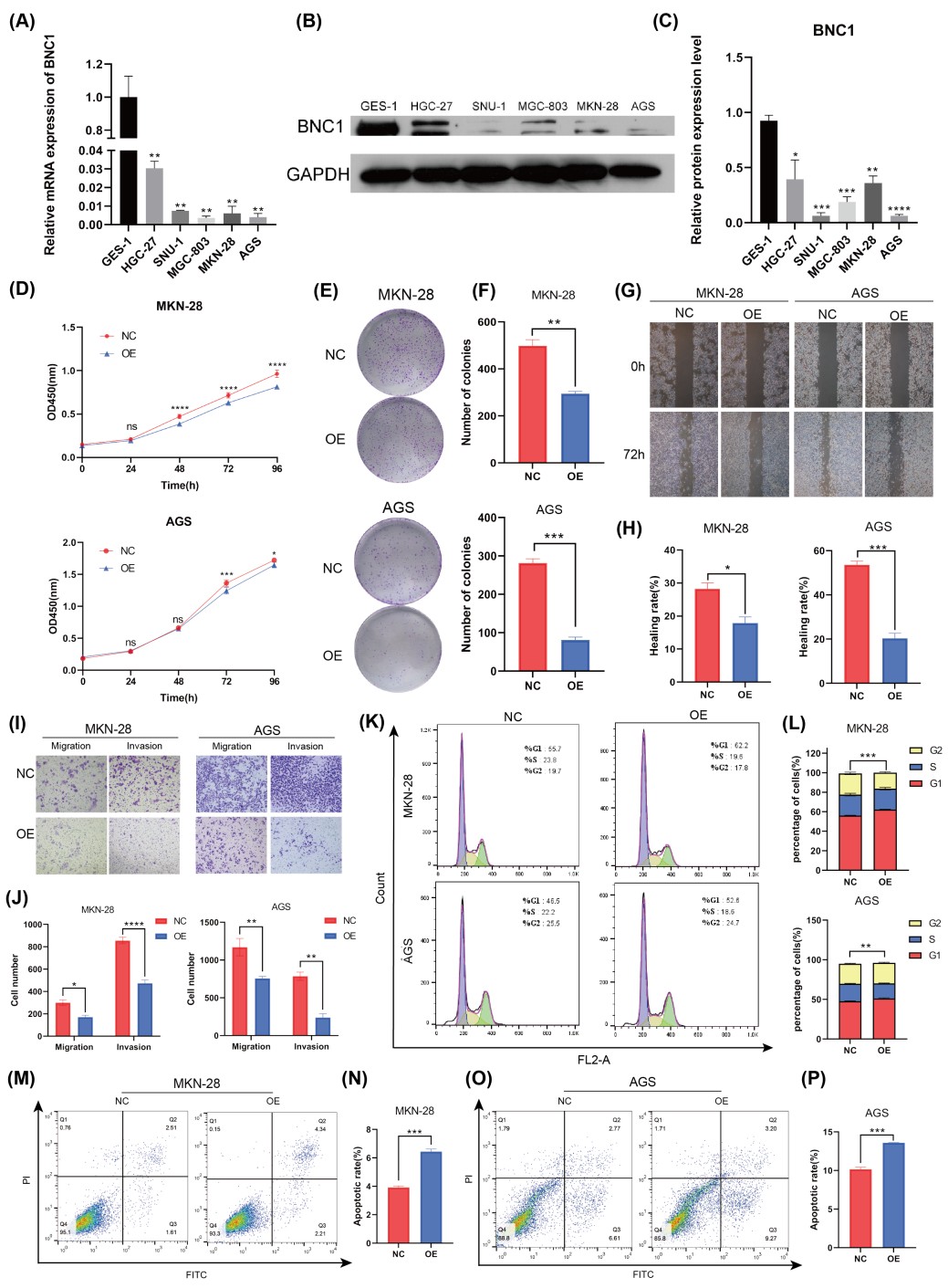

**Figure 2** **BNC1 is a tumor suppressor transcription factor in GC cells.** (A) mRNA levels of BNC1 in normal gastric epithelial cells (GES-1) and gastric cancer cell lines (HGC-27, SNU-1, MGC-803, MKN-28, AGS). (B–C) Relative BNC1 protein expression in normal gastric epithelial cells GES-1 and gastric cancer cells. (D) CCK-8 assay of cell growth with BNC1 overexpression and control cells. (E) Colony formation assay and (F) quantitative analysis of BNC1 overexpression and (continued on next page...)

**Figure 2 (...continued)**
control cells. (G) Wound healing assay and (H) quantitative analysis of BNC1 overexpression and control cells. (I) Migration and invasion assays and (J) quantitative analysis of BNC1 overexpression and control cells. (K–L) Cell cycle was examined by flow cytometric analysis. (M, O) Apoptosis assay and (N, P) quantitative analysis of BNC1 overexpression and control cells.

## BNC1 negatively regulates CCL20

Transcription factors specifically bind to DNA sequences to regulate gene expression. To identify downstream target genes of BNC1, AGS stable cell lines overexpressing BNC1 were established. Total RNA was extracted, purified, and used to construct libraries for sequencing on the Illumina platform. Transcriptome sequencing revealed a predominant downregulation of gene expression upon BNC1 overexpression, suggesting a significant negative regulatory effect. Clustering analysis further underscored this trend (Fig. 4A). Applying criteria of $\log_2 FC < -1.5$ and $P < 0.05$ to the transcriptome data identified 56 downregulated genes. To refine the search for key target genes, the GEPIA2 online bioinformatics tool (*Tang et al., 2019*) was used to download 3,742 highly expressed genes in gastric cancer from The Cancer Genome Atlas (TCGA) database. The intersection of these two gene sets yielded five key genes, as illustrated in the volcano plot (Figs. 4B, 4C). The qPCR validation showed that PTPRC, SERPINB9, and RAC2 had low expression in gastric cancer cells (Figs. 4D–4F), whereas only CCL20 and BANK1 were highly expressed (Figs. 4G, 4H). Consequently, further investigation focused on CCL20 and BANK1. Kyoto Encyclopedia of Genes and Genomes (KEGG) enrichment analysis revealed significant enrichment in the chemokine signaling pathway (Fig. 4I), suggesting that CCL20 is a downstream target of BNC1. Subsequent qPCR and ELISA assays confirmed that overexpression of BNC1 led to downregulation of CCL20 at both mRNA and protein levels (Figs. 4J, 4K). Due to the absence of BNC1 in the JASPAR database (https://jaspar.elixir.no/tfbs_extraction/), the closely related BNC2 was used to predict potential binding sites on the CCL20 promoter (Table S4). The AlphaFold3 predictions suggested a high likelihood of interaction between BNC1 and these sites, implying that BNC1 acts as a transcriptional repressor of CCL20 (Fig. 4L). This hypothesis was validated by dual-luciferase reporter assays, which demonstrated that BNC1 negatively regulates CCL20 promoter activity (Fig. 4M). These findings indicate that BNC1 negatively regulates CCL20 expression in gastric cancer cells.

## CCL20 promotes development of gastric cancer cells

This study established that BNC1 functions as a tumor suppressor in gastric cancer cells and negatively regulates its target gene, CCL20. A series of experiments was conducted to elucidate the biological role of CCL20 in GC progression. SiRNA technology was used to effectively silence CCL20 expression in gastric cancer cell lines (Figs. S3A, S3B). The downregulation of CCL20 resulted in a significant reduction in cell proliferation as determined by CCK8 and Colony Formation assays (Figs. 5A–5C). Additionally, wound healing and Transwell invasion assays demonstrated that CCL20 knockdown markedly suppressed the migratory and invasive capacities of gastric cancer cells compared to controls (Figs. 5D–5G). IHC staining analysis of gastric cancer tissue samples ($n = 37$) revealed that

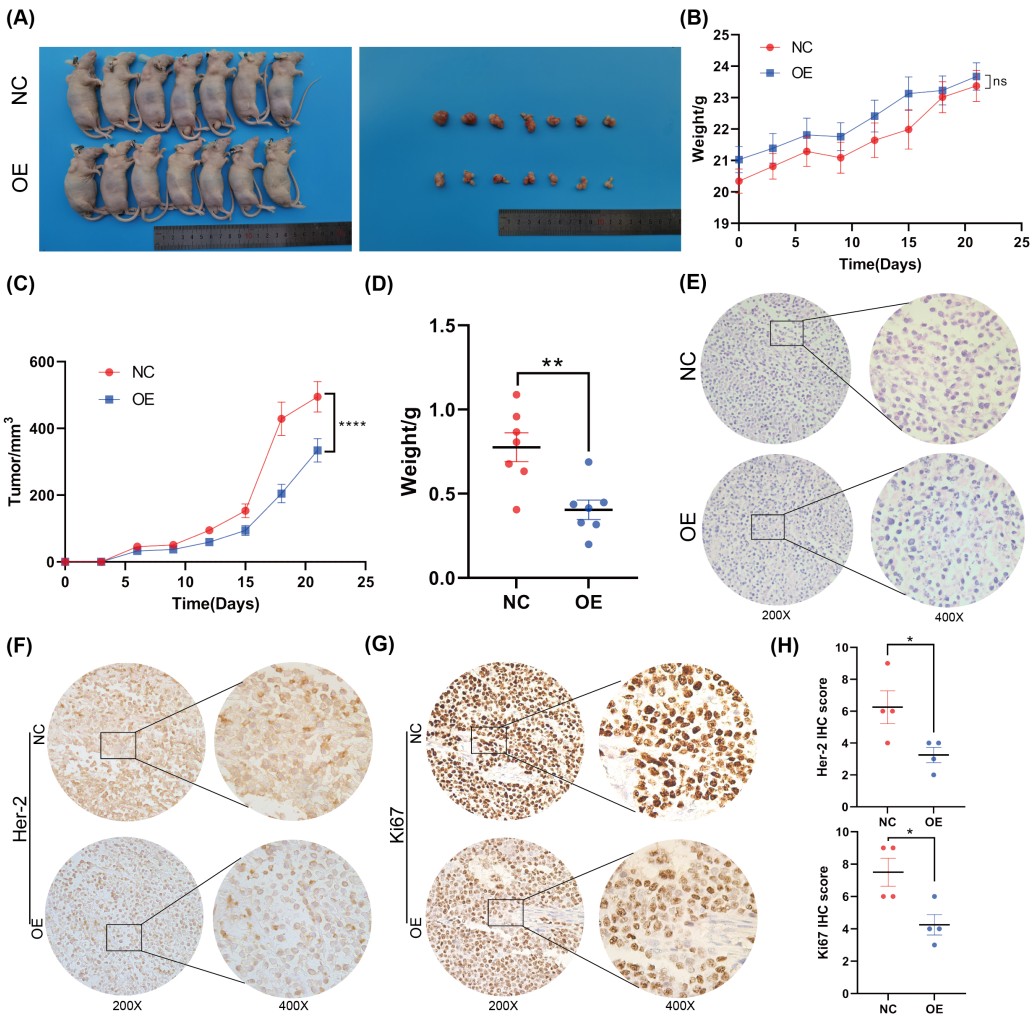

**Figure 3** **Overexpression of BNC1 inhibited the proliferation and metastasis of GC cells *in vivo*.** (A) Images of tumors on day 21. (B) Nude mouse weight growth curves of xenografts generated by BNC1 stable overexpression in AGS cells and control cells ($n = 7$). (C) Tumor growth curves of xenografts generated by BNC1 stable overexpression in AGS cells and control cells. (D) Weight of tumors from each group. (E) Representative H&E staining imaging of tumors (magnification 200× and 400×). (F–G) Representative images from IHC analysis of tumor samples stained with Her-2 and Ki67 ($n = 4$; magnification 200× and 400×) along with (H) staining scores of Her-2 and Ki67.

CCL20 is highly expressed in tumor tissues relative to adjacent normal tissues (Figs. 5H, 5I). Additional analyses using patient clinical information showed a strong correlation between elevated CCL20 expression and lymph node metastasis (Figs. 5J, 5K, and Table S5). No significant associations were observed between CCL20 expression and pathological stage, degree of differentiation, or patient gender (Figs. S3C–S3E). These results indicate that the oncogenic gene CCL20 plays an important role in the biological function of GC cells.

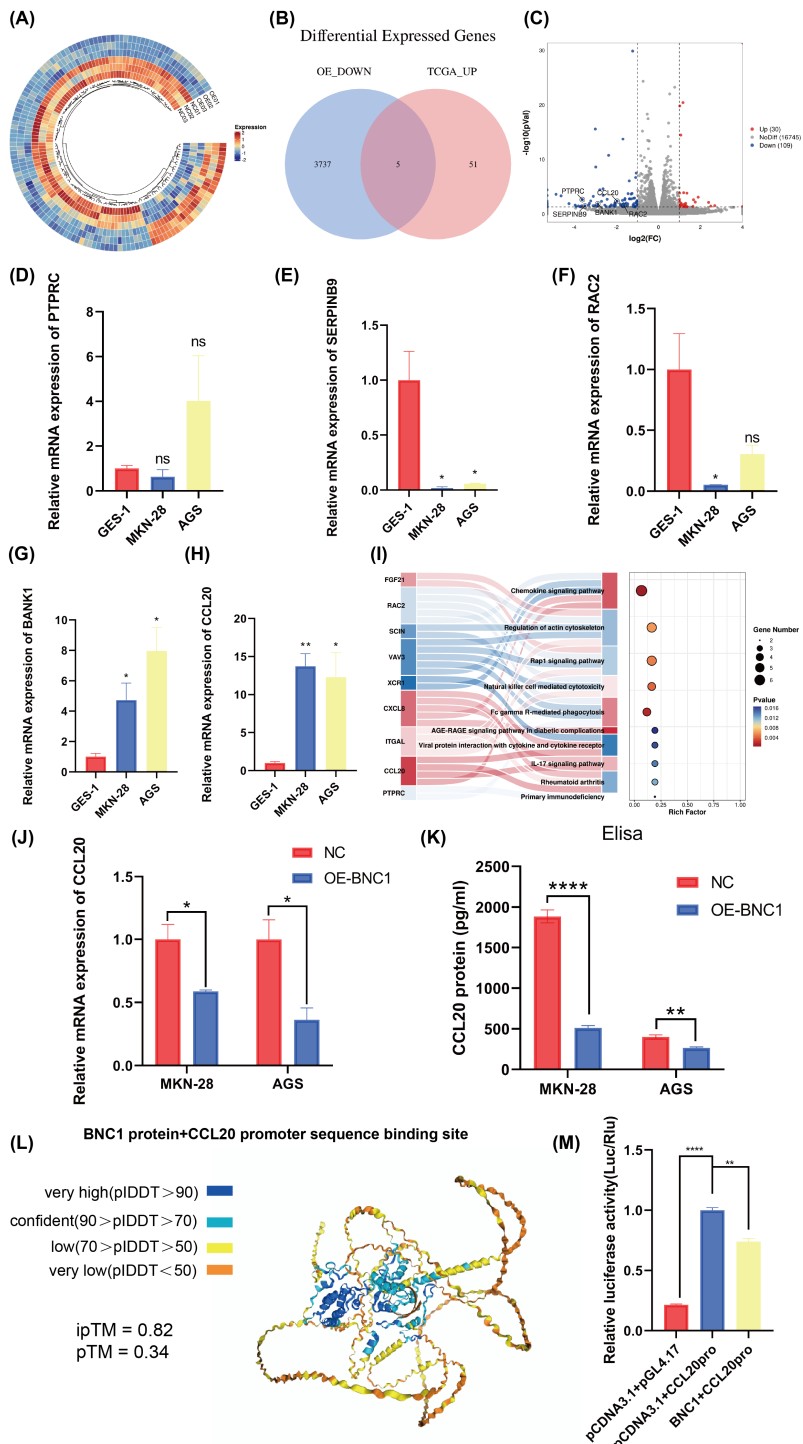

**Figure 4  Screening for downstream target genes of BNC1.** (A) Heatmap of gene expression changes after overexpression of BNC1. (B) Venn diagram showing the screening of target genes of BNC1. Genes showing decreased expression after overexpression of BNC1 (continued on next page...)

**Figure 4 (…continued)**
and highly expressed genes in gastric cancer from the TCGA database. (C) Downstream target genes of BNC1 were screened by transcriptome sequencing in AGS stable cell lines overexpressing BNC1 and control cells (log$_2$ fold change [FC] > 1.5 or < −1.5). (D–H) mRNA levels of PTPRC, SERPINB9, RAC2, BAKN1, and CCL20 in MKN-28, AGS and GES-1 cell lines. (I) KEGG analysis of genes showing decreased expression after overexpression of BNC1. (J) mRNA levels of CCL20 in MKN-28 and AGS stable cell lines overexpressing BNC1 and in control cells. (K) ELISA shows the protein levels of CCL20 in MKN-28 and AGS cell lines after overexpression of BNC1. (L) AlphaFold3 prediction of BNC1 binding to the CCL20 promoter region. (M) Dual luciferase results for BNC1 and CCL20.

### BNC1 acts on the CCL20 promoter to mediate the JAK-STAT signaling pathway to promote apoptosis in gastric cancer cells

KEGG analysis of transcriptome sequencing data revealed significant enrichment of the chemokine signaling pathway in gastric cancer samples (Fig. 4I). Previous studies have shown that CCL20 mediates the JAK-STAT signaling pathway, influencing migration and invasion in breast cancer cells (*Muscella, Vetrugno & Marsigliante, 2017*), and that berberine inhibits apoptosis in gastric cancer *via* the JAK-STAT pathway (*Xu et al., 2022*). In the current study, initial functional assays indicated that overexpression of BNC1 promotes apoptosis in gastric cancer cells (Figs. 2M–2P). These findings suggest that BNC1 acts on the CCL20 promoter to mediate the JAK-STAT signaling pathway, thereby promoting apoptosis. Consistent with this hypothesis, overexpression of BNC1 or knockdown of CCL20 led to decreased expression of JAK2 and inhibited phosphorylation of STAT3 (Figs. 6A–6D). This suppression resulted in diminished expression of the anti-apoptotic protein BCL-2 (Fig. 6E) and an increased expression of the pro-apoptotic protein BAX (Fig. 6F). These results suggest that BNC1 acts on the CCL20 promoter to mediate the molecular mechanism by which JAK-STAT signaling promotes apoptosis.

## DISCUSSION

Accumulating evidence indicates that BNC1 plays a significant role in tumor development. BNC1 is markedly upregulated in bladder squamous cell carcinoma, implying an oncogenic role (*Hurst et al., 2022*). Conversely, BNC1 expression is downregulated in ovarian cancer and pancreatic cancer, suggesting a tumor suppressor function (*Eissa et al., 2019*; *Liang et al., 2022*). This dichotomy may be attributable to its many regulatory targets (*Pangeni et al., 2015*), leading to varying functions in different tumor types. Previous research has shown that BNC1 serves as a key transcriptional activator in esophageal squamous cell carcinoma and that LINC01305 recruits BNC1 to act on G-protein pathway suppressor 1 to promote tumor progression (*Xiong et al., 2023*). However, the role of BNC1 in gastric cancer remains poorly understood. To assess its potential clinical significance, BNC1 expression in gastric cancer tissues was first evaluated using immunohistochemistry. The IHC results showed that BNC1 was significantly downregulated in gastric cancer and that this downregulation correlated with advanced pathological stages and lymph node metastasis. MGC-803 did not exhibit significant biological changes on BNC1 modulation. Previous research suggests that this is because MGC-803 is not a pure clonal gastric cancer cell line but is, in fact, a hybrid cell line, likely resulting from contamination with HeLa cells

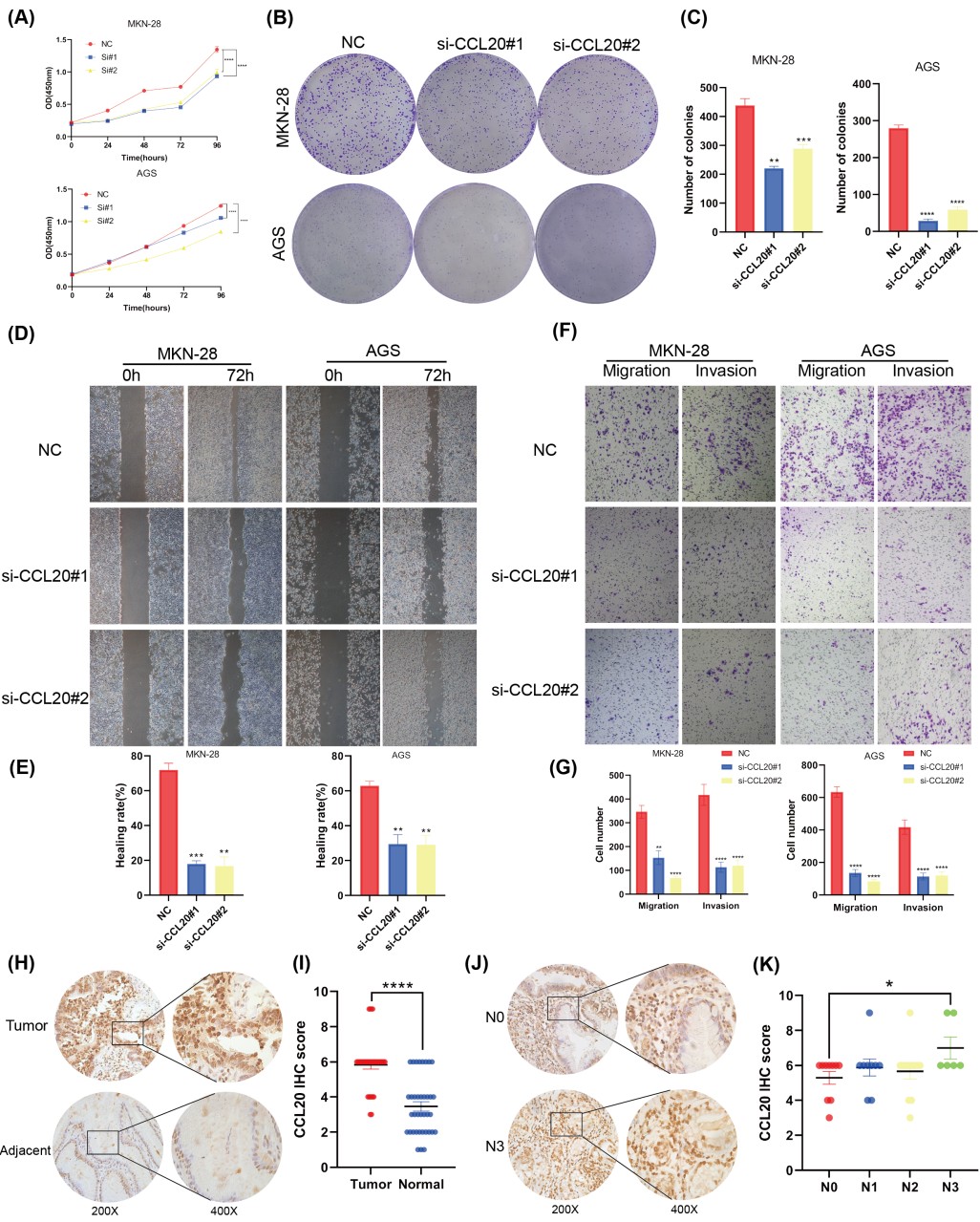

**Figure 5** **CCL20 promotes the development of gastric GC cells.** (A) CCK-8 assay of cell growth with CCL20 knockdown and control cells. (B) Colony formation assay and (C) quantitative analysis of CCL20 knockdown and control cells. (D) Wound healing assay and (E) quantitative analysis of CCL20 knockdown and control cells. (F) Migration and invasion assays and (G) quantitative analysis of CCL20 knockdown and control cells. (H) Representative images of CCL20 IHC staining in gastric tumor tissues and adjacent normal tissues, along with (I) quantification of CCL20 scores ($n = 37$; magnification 200× and 400×). (J) Representative images of CCL20 IHC staining in N0 GC tissues and N3 GC tissues, along with (K) quantification of CCL20 scores ($n = 37$; magnification 200× and 400×).

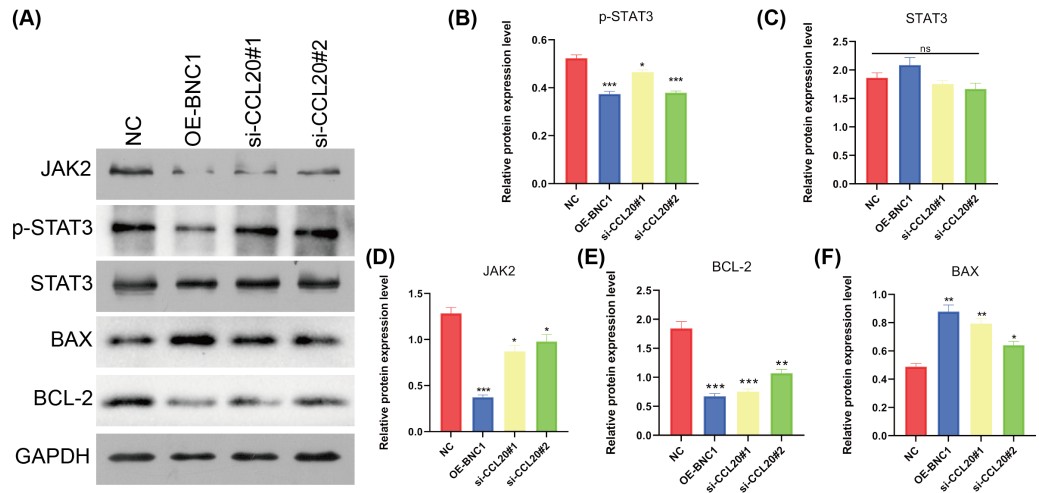

**Figure 6** **BNC1 acts on the CCL20 promoter to mediate the JAK-STAT signaling pathway to promote apoptosis.** (A–F) Western blotting shows the protein levels of JAK2, STAT3, p-STAT3, BAX, and BCL-2 after overexpression of BNC1 or knockdown of CCL20 in MKN-28 cells. NC indicates control.

or a cell fusion event (*Yang et al., 2024*). This mixed cellular origin could have contributed to the lack of significant biological effects in the cell function assays following BNC1 overexpression. This finding highlights the potential for different tumor types to exhibit varying responses to BNC1 modulation depending on the specific characteristics of the cell line used. As a result, the role of BNC1 in gastric cancer was further investigated using more authentic gastric cancer cell lines, such as MKN-28 and AGS, to ensure more reliable and interpretable results. Further functional assays, both *in vitro* and *in vivo*, demonstrated that BNC1 overexpression inhibited tumor cell proliferation and metastasis. These findings suggest that BNC1 may function as a transcriptional repressor in gastric cancer.

Though transcription factors are known to regular gene transcription in various ways (*Karin, 1990*; *Latchman, 1993*; *Lambert et al., 2018*), the regulatory role of the transcriptional repressor BNC1 in gastric cancer has not been previously explored. It is well-established that these repressors often exert their effects by binding to the transcription initiation site or to adjacent sequences, thereby inhibiting the formation of the transcription initiation complex (*Francois, Donovan & Fontaine, 2020*). Given this classical mechanism, BNC1 may function in a similar manner to mediate its repressive effects in gastric cancer. To identify BNC1's potential downstream factors, transcriptome sequencing was performed on GC cells overexpressing BNC1. The results showed that BNC1 predominantly downregulates gene expression. Integration of this data with the TCGA database revealed several candidate targets including CCL20, BANK1, PTPRC, SERPINB9, and RAC2. The qPCR validation confirmed that CCL20 and BANK1 were significantly upregulated in GC. KEGG enrichment analysis showed significant enrichment of the Chemokine signaling pathway, suggesting a possible regulatory link between BNC1 and CCL20. Further experiments demonstrated that overexpression of BNC1 resulted in a significant downregulation of CCL20 expression. Dual-luciferase reporter assays

confirmed that BNC1 directly binds to the CCL20 promoter, suppressing its transcription. These results suggest that CCL20 is a negative regulatory target of BNC1 in gastric cancer.

Cytokines, a class of proteins produced by various cell types, play essential roles in modulating immune responses and facilitating intercellular communication (*O'Shea & Murray, 2008*; *Kulbe et al., 2012*). They are categorized into interleukins, interferons, tumor necrosis factors, growth factors, hematopoietic factors, and chemokines (*Zlotnik & Yoshie, 2012*). CCL20, also known as macrophage inflammatory protein-3$\alpha$ (MIP-3$\alpha$), Exodus-1, or liver and activation-regulated chemokine (LARC), is a multifunctional chemokine widely expressed in human tissues and immune cells, and is particularly abundant in the lymph nodes, lungs, and liver (*Hieshima et al., 1997*; *Power et al., 1997*; *Xiao et al., 2015*). Previous studies have shown that CCL20 induces migration and invasion of gastric cancer cells (*Han et al., 2015*). In lung cancer, CCL20 is highly expressed and promotes cell migration and proliferation *via* autocrine signaling (*Wang et al., 2016*), highlighting its oncogenic role in tumor progression. Through a series of cellular biological experiments and analyses of clinical sample data, CCL20 was confirmed to be highly expressed in gastric cancer tissues. These data also demonstrated CCL20's role in enhancing the proliferation and migration of gastric cancer cells *in vitro*. These findings implicate the BNC1-CCL20 axis as a critical pathway in gastric cancer progression and highlight CCL20 as a potential therapeutic target for intervention.

After identifying CCL20 as a direct target of BNC1, transcriptome sequencing was used to explore the downstream signaling pathways regulated by BNC1-CCL20. KEGG analysis revealed that BNC1 overexpression significantly enriched chemokine signaling pathways. CCL20, a chemokine involved in the JAK-STAT signaling pathway, has been shown to influence cell migration and invasion in breast cancer (*Muscella, Vetrugno & Marsigliante, 2017*). The JAK-STAT signaling cascade, influenced by various cytokines, regulates key cellular processes including cell differentiation, metabolism, survival, homeostasis, multidrug resistance, and immune responses (*Ihle et al., 1994*; *Ihle & Kerr, 1995*; *O'Shea, Gadina & Schreiber, 2002*; *Laurino et al., 2023*; *Ding et al., 2025*). In gastric cancer, berberine inhibits cell apoptosis *via* the JAK-STAT pathway (*Xu et al., 2022*). Additionally, IGFBP7 regulates cell proliferation and migration in gastric cancer through the JAK-STAT signaling pathway (*Mo et al., 2024*). The initial functional assays from this study demonstrated that BNC1 overexpression promotes gastric cancer cell apoptosis. CCL20 mediates its biological effects through specific binding to its receptor, CCR6, a G-protein-coupled receptor (*Baba et al., 1997*; *Han et al., 2015*; *Kadomoto, Izumi & Mizokami, 2020*). These findings suggest that BNC1, through transcriptional regulation of CCL20, mediates JAK-STAT signaling to promote gastric cancer cell apoptosis. Overexpression of BNC1 or knockout of CCL20 decreased the expression levels of JAK2, phosphorylated STAT3, and BCL-2, while increasing BAX expression.

In summary, these findings suggest that BNC1 may regulate CCL20 expression in gastric cancer cells through direct binding to the CCL20 promoter region, inhibiting its transcriptional activity. This repression of CCL20 expression consequently leads to a reduction in the levels of its downstream target, JAK2. The downregulation of JAK2 further suppresses the phosphorylation of STAT3, a key event in the JAK-STAT signaling

pathway. This inhibition of JAK-STAT signaling ultimately results in the suppression of gastric cancer cell proliferation, migration, and invasion, while promoting apoptosis. These mechanisms collectively highlight BNC1's potential as a tumor suppressor in gastric cancer, with significant implications for the development of novel therapeutic strategies targeting the CCL20/JAK-STAT axis.

Despite the insights gained, this study has certain limitations. Relying solely on transcriptomic analysis for target screening may not precisely identify the functional molecules involved, and integrating multiple experimental approaches could improve molecular localization. Although the tumor microenvironment (TME) is critical in cancer biology, this study did not extensively explore the impact of the CCL20/CCR6 axis on the immune TME. However, transcription factors rarely function in isolation, as transcriptional repressors may competitively bind to sites overlapping with activators or interact with known activators at distinct sequences (*Reynolds, O'Shaughnessy & Hendrich, 2013*; *Chen et al., 2022*). These limitations highlight the need for further research and may become a focus of subsequent studies.

## CONCLUSIONS

In conclusion, this study demonstrates that BNC1 inhibits the development and progression of gastric cancer cells by acting on the CCL20 promoter to mediate the JAK-STAT signaling pathway (Fig. 7). Low BNC1 expression correlates with advanced pathological stage and lymph node metastasis, while high CCL20 expression is associated with lymph node metastasis in gastric cancer patients. These findings may offer new therapeutic strategies for treating gastric cancer.

## ACKNOWLEDGEMENTS

The authors thank the members of the laboratories at the Institute of Tissue Engineering and Stem Cells, Nanchong Central Hospital, for technical assistance and discussion. We extend our heartfelt gratitude to the Affiliated Hospital of North Sichuan Medical College for their generous support in collecting clinical samples, which significantly contributed to this study. We are also deeply thankful to the Innovation Center for Science and Technology of North Sichuan Medical College for providing the experimental platform and technical support. Furthermore, we acknowledge the authors and contributors of The Cancer Genome Atlas (TCGA) database for their invaluable efforts in generating and sharing high-quality sample data, which served as a vital resource for our research. The authors acknowledge the use of ChatGPT (OpenAI) for language polishing. The authors reviewed and revised the generated text to ensure accuracy and alignment with the scientific content.

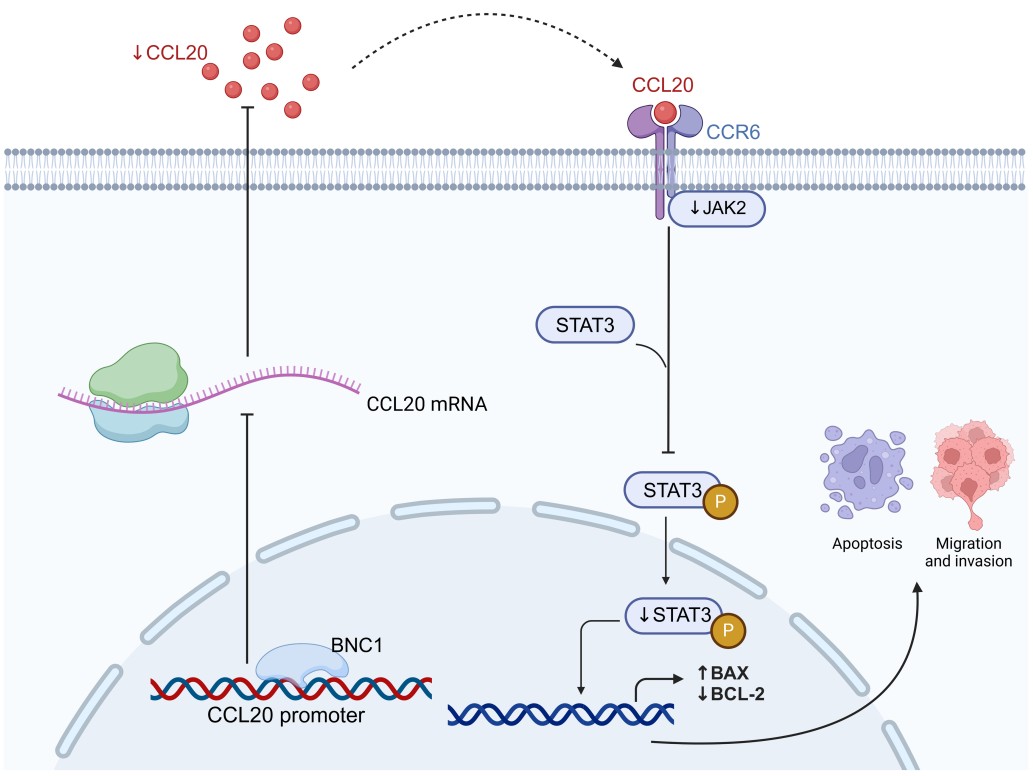

**Figure 7** **BNC1 acts on the CCL20 promoter to suppress the expression of CCL20 in GC cells.** CCL20 expression mediates the JAK-STAT signaling pathway, leading to the development and progression of GC cells. Created with BioRender.com.

### Funding

The research was funded by the Sichuan Provincial Department of Science and Natural Science Foundation of Sichuan Province of China, with the project number: 2024NSFSC0737. The research was also funded by Sichuan Provincial Administration of Traditional Chinese Medicine Subjects, with the project number: 2024MS590. Additionally, the research was supported by a project from the Leshan Municipal Health Commission, with the project number: 23SZD019. The funders had no role in study design, data collection and analysis, decision to publish, or preparation of the manuscript.

### Grant Disclosures

The following grant information was disclosed by the authors:
The Sichuan Provincial Department of Science and Natural Science Foundation of Sichuan Province of China: 2024NSFSC0737.
Sichuan Provincial Administration of Traditional Chinese Medicine Subjects: 2024MS590.
Leshan Municipal Health Commission:  23SZD019.

## Competing Interests

The authors declare there are no competing interests.

## Author Contributions

- Lixin Liu conceived and designed the experiments, performed the experiments, analyzed the data, prepared figures and/or tables, authored or reviewed drafts of the article, and approved the final draft.
- Li Xiong analyzed the data, authored or reviewed drafts of the article, and approved the final draft.
- Hong Peng analyzed the data, authored or reviewed drafts of the article, and approved the final draft.
- Qin Deng analyzed the data, prepared figures and/or tables, and approved the final draft.
- Kang Liu conceived and designed the experiments, authored or reviewed drafts of the article, and approved the final draft.
- Shusen Xia conceived and designed the experiments, authored or reviewed drafts of the article, funding acquisition, and approved the final draft.

## Human Ethics

The following information was supplied relating to ethical approvals (i.e., approving body and any reference numbers):

The collection and use of patient samples were approved by the ethics committee of the Affiliated Hospital of the North Sichuan Medical College.

## Animal Ethics

The following information was supplied relating to ethical approvals (i.e., approving body and any reference numbers):

Animal studies were carried out according to the protocols approved by the ethics committee of the Institutional Animal Care and Use Committee of the North Sichuan Medical College.

## Data Availability

The raw data from flow cytometry experiments conducted in this study are available at Figshare: Liu, Lixin (2024). Supplemental Files. figshare. Dataset. https://doi.org/10.6084/m9.figshare.28051373.v2

The original, unprocessed Western blot images are available in the Supplementary Files.

## Supplemental Information

Supplemental information for this article can be found online at http://dx.doi.org/10.7717/peerj.19477#supplemental-information.

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
