# Peer review of "BNC1 inhibits the development and progression of gastric cancer by regulating the CCL20/JAK-STAT axis"

_PeerJ, doi:10.7717/peerj.19477_

## Round 0.1 · original submission · Minor Revisions

Address comments of all reviewers and provide response in a point by point manner.

Reviewer 1 ·

Basic reporting

1. Some sentences are unnecessarily long and complicated, which can hinder readability. For instance:
- Original: "BNC1 overexpression significantly inhibited cell proliferation compared to control cells (Fig. 2D-F)."
- Suggested Change: "Compared to control cells, BNC1 overexpression significantly inhibited cell proliferation (Fig. 2D-F)."

2.There are occasional spelling errors, such as "magnification" being mistakenly written as "magniûcation."

3.While the paper frequently mentions BNC1's tumor-suppressing role in gastric cancer cells, it lacks a detailed explanation of the underlying mechanism. In particular, the discussion on BNC1’s regulation of CCL20 and its effect on the JAK-STAT signaling pathway could be more coherent. - For example, when discussing how BNC1 regulates CCL20, there should be a more detailed explanation of how BNC1 binds to the CCL20 promoter and how this influences the JAK-STAT pathway.

4.The figure legends are occasionally too brief, and key experimental results lack sufficient explanation. For instance, in Figure 1, the BNC1 immunohistochemistry results could be more clearly tied to their clinical significance.

5. Although the paper mentions BNC1's role in various cancers, its specific role in gastric cancer remains underexplored. The discussion could benefit from a more in-depth explanation of BNC1 as a transcriptional repressor.

6. Please consider including the following references in this article.
(1) Gao S, Li J, Wang W, Wang Y, Shan Y, Tan H. Rabdosia rubescens (Hemsl.) H. Hara: A potent anti-tumor herbal remedy - Botany, phytochemistry, and clinical applications and insights. J Ethnopharmacol. 2025;340:119200. doi: 10.1016/j.jep.2024.119200. Epub ahead of print. PMID: 39631716.
(2) Eissa MAL, Lerner L, Abdelfatah E, Shankar N, Canner JK, Hasan NM, Yaghoobi V, Huang B, Kerner Z, Takaesu F, Wolfgang C, Kwak R, Ruiz M, Tam M, Pisanic TR 2nd, Iacobuzio-Donahue CA, Hruban RH, He J, Wang TH, Wood LD, Sharma A, Ahuja N. Promoter methylation of ADAMTS1 and BNC1 as potential biomarkers for early detection of pancreatic cancer in blood. Clin Epigenetics. 2019 Apr 5;11(1):59. doi: 10.1186/s13148-019-0650-0. PMID: 30953539; PMCID: PMC6451253.
(3)Shiyong Gao, Yanmin Shan, Yue Wang, Weiya Wang, Jianwen Li, Huixin Tan. Polysaccharides from Lonicera japonica Thunb.: Extraction, purification, structural features and biological activities—A review. International Journal of Biological Macromolecules. 2024 Nov; 281P4 136472. https://doi.org/10.1016/j.ijbiomac.2024.136472
(4) Mo W, Deng L, Cheng Y, Ge S, Wang J. IGFBP7 regulates cell proliferation and migration through JAK/STAT pathway in gastric cancer and is regulated by DNA and RNA methylation. J Cell Mol Med. 2024 Oct;28(19):e70080. doi: 10.1111/jcmm.70080. PMID: 39351597; PMCID: PMC11443158.
(5) Han G, Wu D, Yang Y, Li Z, Zhang J, Li C. CrkL meditates CCL20/CCR6-induced EMT in gastric cancer. Cytokine. 2015 Dec;76(2):163-169. doi: 10.1016/j.cyto.2015.05.009. Epub 2015 Jun 1. PMID: 26044596.

Experimental design

no comment

Validity of the findings

no comment

Additional comments

no comment

Reviewer 2 ·

Basic reporting

Liu et al., highlighted the pivotal role of BNC1 as suppressor of GC development and progression trough CCL20/JAK-STAT Axis. Manuscript appear well designed and clearly written. However, authors should check for English and minor typo.
Introduction and discussion sections could benefit of the addition of recent literature regarding the influence of cytokines on the JAK-STAT signaling cascade, (e.g. PMID: 36979673, PMID: 39490801, PMID: 39129286).

Experimental design

The experimental design rationale appear solid. Figure legend should be improved (e.g. Abbreviation should be exploited).

Validity of the findings

Authors clarified the role of BNC1 and CCL20 in the pathogenesis of gastric cancer. The sentence at lines 333-334, is not clear. Are previous studied referred to this work? In this case, "previous" should be modified in "our".
Conclusion are supported by the findings.

Additional comments

no comment

Reviewer 3 ·

Basic reporting

The english is mostly clear and looks professional.
A only have a couple of basic comments on the language used:
1. Space between a word and a bracket should be consistent. For example. Line 39.. worldwide[SPACE](Bray et al., 2024).
2. I recommend rephrasing this on line 45: identifying effective transcription factor targets is therefore crucial for developing novel therapeutic strategies for gastric cancer. Calling transcription factor target effective does not look professional.
3. I also recommend rephrasing this line 49: Initially identified in vitro within cultured human epidermal keratinocyte. I would not use the word “within”

Experimental design

Yes, the research is within the aims and scope of PeerJ. The research question is also well defined, and fills a knowledge gap. The methods are described correctly.

Validity of the findings

Validity seems good, however i could see raw data files for the proliferation and IHC. Not sure if that is necessary here.
One point is on the Immunohistochemistry methods: Must make clear that if the scoring was done in a blinded manner or not. Blinded scoring substantially increases confidence in the data.
Fig2D-F: The authors should specify what is the lentiviral control used here and update the methods.

Additional comments

Some other comments on experimental design
4. Fig1: Since the authors are claiming expression of BNC1 leads to these effects- Could the authors use TCGA datasets (or similar) to prove that BNC1 expression correlates to pathological stage in Gastric cancer? At least the authors must confirm the expression at the RNA level in patient samples (or give a contrary explanation)
5. Fig2D and fig S2: I would recommend the authors to use actual cell numbers here- or using optical density to extrapolate actual cell numbers (rather than just giving optical density numbers). The difference in cell growth seems tiny even after 96 hours does raise some concerns about the validity of this assay.
6. I do appreciate the authors acknowledging in the manuscript that modulation of BNC1 in MGC-803 cell line did not result in significant biological changes. Would be great to get some explanation or hypothesis on that front.
7. Fig2D-F: The authors should specify what is the lentiviral control used here and update the methods.
8. Fig3: Was there any difference in survival in xenograft experiments? Trying to understand if these differences are “significant” enough for clinical relevance.
9. Fig4: Chromatin immunoprecipitation (or ChIP) experiments confirming the BNC1 at the promotor of CCL20 could really help strengthen the mechanism.
10. Fig S3C-E: They authors show that no significant associations were observed between CCL20 expression and pathological stage, degree of differentiation, or patient gender. This does create some doubt in my mind that the proposed mechanism of BNC1 controlling the expression CCL20 which is leading to pathogenesis is real or not. The authors should offer some explanations on that front.
a. Does overexpressing both BNC1 and CCL20 negate the effects of BNC1 overexpression?

Annotated reviews are not available for download in order to protect the identity of reviewers who chose to remain anonymous.

·

Basic reporting

1. Basic reporting - the article (throughout the text) uses clear, unambiguous and professional English, the Introduction and background to show context, Literature well referenced and relevant structure confirms to PeerJ standards, discipline norm and improved for clarity, Figures (1 to7) are relevant, high quality, well labelled and described, Methods described with sufficient detail. All underlying data have been provided, they are robust and statistical sound.

Experimental design

2. Experimental design – Research question well defined, relevant and meaningful. The investigation performers to a high technical and ethical standard. The Methods described with sufficient detail and information to replicate. Conclusions are well stated, linked to oryginal reseaerch question.

Validity of the findings

3. Validity of the findings - BNC1 inhibits the development and progression of gastric cancer cells by acting on CCL20 to mediate the JAK-STAT signaling pathway. Low BNC1 expression correlates with advanced pathological stage and lymph node metastasis. Studies have shown that overexpression of BNC1 promotes apoptosis of gastric cancer cells.

Additional comments

General comments - interesting publication, searching for new therapeutic strategies for patients with gastric cancer.

---

## Round 0.2 · accepted · Accept

Authors have addressed all of the reviewers' comments and manuscript is ready for publication.

Reviewer 2 ·

Basic reporting

no comment

Experimental design

no comment

Validity of the findings

no comment

Additional comments

Authors followed the suggestions by implementing introduction, discussion and figure legends enhancing the quality of the work.